# Objective Sustainability Assessment in the Digital Economy: An Information Entropy Measure of Transparency in Corporate Sustainability Reporting

Mohammed Zakaria [1], Chadi Aoun [1] and Divakaran Liginlal [2,*]

1   Dietrich College of Humanities and Social Sciences, Carnegie Mellon University, Pittsburgh, PA 15289, USA; mozamaruf@gmail.com (M.Z.); chadi@cmu.edu (C.A.)
2   Dietrich College of Humanities and Social Sciences & Heinz College of Information Systems, Carnegie Mellon University, Pittsburgh, PA 15289, USA
*   Correspondence: liginlal@cmu.edu; Tel.: +1-412-818-2884

**Abstract:** The Internet is now a central enabler for sharing sustainability information. Yet, such enablement is complicated through an exponentially increasing array of information. What is lacking in the digital economy are objective and transparent mechanisms to provide reliable assessments of the published sustainability information in a timely and efficient manner. In addressing such limitation, this research proposes an objective automated mechanism for measuring transparency in sustainability reporting using an information entropy-based approach. Through text-mining methods and expert validation, the study built a sustainability dictionary corpus and then applied the corpus for objectively assessing the relative entropy between the probability distributions of words in the sustainability dictionary and those in corporate reports. To demonstrate its effectiveness, the mechanism was empirically applied to compare sustainability reporting of organizations in the energy sector. Here, the research effectively compared cartels with non-cartels by assessing the sustainability reports of major OPEC (Organization of the Petroleum Exporting Countries) and non-OPEC producers spanning a three-year period and found consistent differences in transparency between the two groups. The findings demonstrate likely normative transparency pressures on disaffiliated producers for which cartels may be immune. The automated mechanism holds important theoretical and practical contributions to the field of sustainability as it provides a rapid and objective means for textual analysis of sustainability information, thus promoting transparency in sustainability reporting in the rapidly evolving digital economy.

**Keywords:** sustainability; corporate transparency; information theory; text analytics; corpus; dictionary; entropy; energy sector

## 1. Introduction

Transparency is an essential pillar for sustainability, as evident in global multilateral environmental agreements. The Paris Agreement [1], in particular, is operationalized through an "enhanced transparency framework" to build mutual trust and promote effective implementation. Such transparency is expected to drive global action, accountability, scrutiny among states and institutions, and engage the global public [2].

With the advent of the digital economy, citizens are increasingly conscious of their rights and demanding more transparency in information [3]. Therefore, transparency is becoming central to business success. Businesses use various communication media like annual reports, social media, and websites to enhance their transparency, and meet public expectations. Such communication often reports on finances, operations, and corporate governance. The past two decades, have introduced a further dimension for corporate transparency—sustainability—which is now required in many jurisdictions, and has led to the publication of annual sustainability reports.

While there are many perspectives on sustainability, there is a common agreement on the fact that it is multidimensional and long term oriented. For instance, Kuhlman and Farrington define sustainability as meeting an objective or goal with economic and societal feasibility through maintaining a constant sum of natural and man-made resources for the foreseeable future [4]. This definition builds upon the Brundtland Commission [5] which defines sustainable development as meeting current needs without compromising on the ability of future generation to meet their own needs.

Sustainability is considered as having three main dimensions: environmental, economic, and social. Environmental sustainability focuses on the maintenance of natural capital and consists of four key activities, which from the source side are the use of renewable and nonrenewable resources, and from the sink side include pollution and waste assimilation [6]. Economic sustainability involves using assets efficiently and generating profits indefinitely [7]. On the other hand, social sustainability has garnered considerably less attention compared with environmental and economic sustainability. It focuses on the impacts of social systems, processes, and structures on people both positively and negatively. In 2010, Forbes claimed that sustainability cannot be achieved until companies are transparent about it because transparency amplifies corporate accountability [8].

The term "transparency" is defined as the availability of information to others. Transparency implies openness, communication, and accountability. In business, corporate transparency focuses on the availability of company-related information to stakeholders and others outside of publicly traded firms [9]. According to Dubbink et al. [10], corporate transparency is gauged by the amount of financial and corporate information to which the public has access. It is crucial for business to be transparent so that stakeholders can use the required information for collaboration and decision-making. Many corporations around the world publish sustainability reports along with other sustainability data and benchmarks. However, no specific sustainability standard or framework exists for these sustainability reports. Therefore, although many corporations provide transparent information about their practices, some may omit or overload their sustainability reports with information that reduces the quality and clarity of their sustainable practices. Moreover, such reports often become susceptible to a range of time-consuming subjective interpretations, often from casual commentators and "pseudo-experts," thus limiting their reliability and relevance. This points to a significant limitation, diminishing the value of sustainability reporting due to the lack of timely and objective assessments.

The goal of this study is to mitigate the aforementioned limitations, by developing an objective automated mechanism for timely assessment of transparency in sustainability reporting, commensurate with stakeholders needs in the digital economy. The study therefore aims to answer the following research questions: (1) What is an effective measure of corporate transparency in sustainability reporting? and (2) How can this measure be applied to rate and compare the organizations in a specific industry sector? In answering these questions, this research study develops an information-theoretical measure of corporate transparency in sustainability reporting and illustrates the effectiveness of such measure by applying it in the energy industry to empirically compare transparency in sustainability reporting between OPEC (Organization of the Petroleum Exporting Countries) and non-OPEC producers based on historical data.

The paper is organized as follows. The next section reviews the extant literature related to transparency in sustainability reporting, corporate transparency measures, and text analytics based on information-theoretical approaches. This is followed by a discussion of the research methodology. The subsequent section discusses the building of the sustainability dictionary and scoring of corporate reports followed by an analysis of the resulting data. The paper concludes with a presentation of the contributions and limitations of this study, and future research avenues.

## 2. Literature Review

### 2.1. Transparency in Sustainability Reporting

In prior studies, researchers have provided a range of definitions for corporate transparency and its implication for organizations. Bushman et al. [9] defined transparency as the availability of firm-related information to the public, which is external to the organization. Although authors like Williams [11] have defined transparency within a complex framework of relevance, time, and reliability, other authors like Bushman et al. [9] have used three other indicators for transparency. These indicators are quality of corporate reporting, intensity of information related to private assets, and information dissemination. Similarly, Dubbink et al. [10] identified three criteria to evaluate transparency: efficiency, freedom, and virtue. Their first criterion highlights the importance of the quality of information while the second and third criteria look at the moral and ethical aspects of reporting. Looking at the importance of transparency in corporate reporting, Dubbink et al. [10] propose that transparency enhances dynamic efficiency and innovation. More specific to our context, transparency is expected to improve sustainable practices by bringing such practices and their associated impact to "sunlight" [2]. This calls for the development of objective measures of transparency to ascertain the effectiveness of corporate sustainability reporting [12], as higher corporate transparency in sustainability reporting plays a key role in action on sustainability.

The need for objective measures is better understood by examining some prominent examples of transparency measures proposed in the literature. Dočekalová and Kocmanová [13] proposed a complex performance indicator (CPI) that integrated the environmental, social, economic, and corporate governance performance of a company. CPI contains seventeen key performance indicators (KPI). Piechocki [14] proposed the transparency scorecard defined by eight factors categorized into linguistic, thematic, and depth indicators. A five-point Likert scale was used to rate the degree of presence of each criterion (total of 50 criteria) and then combined using a heuristic method. Schnackenberg [15] used a game-theoretical approach to measuring transparency based on three criteria—disclosure, clarity, and accuracy. Standard and Poor's Transparency & Disclosure Score is primarily based on the quantity of information disclosure and does not take into account quality of information [16]. Lee and Saen [17] introduced a data envelopment analysis (DEA) technique (linear programming approach) based on three key principles (economic transparency and profitability, social responsibility, environmental sustainability) and five key areas of corporate sustainability performance (governance, accountability, human rights, social contribution, and environmental management and innovation). Yet another measure proposed is the sustainability balanced scorecard (SBSC) which is based on strategic objectives formulated along these perspectives—financial, customer, processes, learning and growth, and non-market [18]. The CSR ranking (https://www.csrhub.com/csrhub/) is another example of a multi-dimensional measure derived from 12 indicators of employee, environment, community and governance performance and relying on secondary data sources. Shahi et al. [19] uses an automated scoring approach to classify sustainability reports by applying machine learning approach to text categorization. Here the GRI environmental subclass and related performance indicators are used as the basis for classification. The researchers used a heuristic scoring approach to classify each corporate document into application levels and compared the resulting scores with those claimed by report authors. The method provides high accuracy in discovering disclosure items but only moderately accurate results in scoring a document. More examples of multidimensional measures that require subjective ranking by experts include those proposed in Delmas and Blass [20] and Eldomiaty [21]. Table 1 summarizes the key characteristics of these seven measures of transparency reported in the extant literature. It is important to note that only the method reported in [19] lends itself to automation although the scoring approach is still subjective. On the other hand, the key goal of the research proposed in Schnackenberg [15] and Lee and Saen [17] was to develop an objective measure, but the underlying game-theoretical and DEA approaches still required point estimates by experts.

| Measure | Dimensions | Reference |
|---|---|---|
| Complex Performance Indicator (CPI) | 4 factors and 17 KPIs | Dočekalová & Kocmanová (2016) [13] |
| Transparency Scorecard | 8 factors and 3 indicators | Piechocki (2004) [14] |
| Standard & Poor's T&D score | 3 categories and 98 items | Patel & Dallas (2002) [16] |
| CSR Rating | 12 indicators | https://www.csrhub.com |
| Sustainability Balanced Scorecard (SBSC) | 5 perspectives | Figge et al. (2002) [18] |
| Game theoretical model | 3 criteria | Schnackenberg (2009) [15] |
| Data Envelopment Analysis (DEA) | 3 key principles and five areas | Lee & Saen (2012) [17] |
| Intelligent Scoring Method | GRI 3.0 environmental subclass and related 30 performance indicators | Shahi et al. (2014) [19] |

Evidently, there are a range of criteria proposed for assessing transparency in sustainability reporting. However, such criteria remain susceptible to subjectivity and delays in rendering assessment because of their time-consuming requirements.

### 2.2. Corporate Sustainability Reporting Practices

Historically, corporate reports served as the main medium for corporations in sharing information about their sustainability performance [22]. According to Hess [23], corporate reports disclose information regarding the sustainability practices and approaches of companies and their performance over a period of time. In other words, corporate reports play a crucial role in providing shareholders with information related to sustainability in any industry. Yet, given the demand for breadth in sustainability impacts, organizations have increasingly published separate sustainability reports.

Corporate sustainability reporting practices have attracted limited research attention. Oncioiu et al. [24] found a dual benefit for sustainability reporting, in promoting the enterprise, and in providing a source of accessible information to customers, investors, and other parties interested in learning about the social and environmental impacts of an enterprise [24]. Hence, such reporting provides a degree of external transparency to stakeholders—building trust, as well as internal productivity impacts on employees, which could result in improved financial performance. Delmas and Blass [20] looked at 15 companies and measured their corporate sustainability reporting by using three sets of indicators (impact, compliance, and management), and noted problems with using single indicators for assessment. On the other hand, Eldomiaty [21] looked at corporate transparency by using three dimensions that include measures of governance structures, a company's competitive position, and the risk of financial transformation. Landrum and Ohsowski [25] used content analysis to categorize a company's sustainability report into one of the five stages of sustainability ranging from weak to very strong. The applied method is questionable as it uses only a normalized count of the frequencies of a set of author-defined keywords corresponding to each stage. Székely and Vom Brocke [26] on the other hand applied topic modeling, specifically Latent Dirichlet Allocation (LDA), to identify 42 topics and related keywords in 9514 corporate reports over the period 1999–2015. Of these, 8 topics were identified related to environmental sustainability with a plethora of words corresponding to general business terms. Li et al. [27] studied the quality of information disclosure in sustainability reports based on six factors—completeness, adequacy, relevance, reliability, normativeness, and clarity and related indicators. Overall, 7 reports were evaluated by 12 experts on a linguistic rating scale for each indicator, with the weight of an indicator determined as a fuzzy entropy (not related to Shannon entropy). Aureli [28] compares the use of content analysis with text mining of corporate reports to identify changes in sustainability-related disclosure by companies aimed at restoring

corporate reputation after an industrial disaster. Change in frequency distribution of a set of keywords or coded elements corresponding to GRI (Global Re-porting Initiative) structural elements was employed as a measure. The results indicate that the two methods are likely to lead to different conclusions. Shin et al. [29] examine key journal articles on sustainability in Maritime studies using text mining techniques. Specifically, the research maps sustainability issues through the LDA topic modeling approach for latent data discovery and relationships among text document data. Besides categorizing topics based on keyword frequencies specific to sustainability in maritime studies, the research also does co-occurrence analysis and represents the results as a dendrogram. Nonetheless, a broader automated quantitative approach to objectively analyze corporate reports and subsequently compare different companies in an industry remains lacking. Nonetheless, a broader automated quantitative approach to objectively analyze corporate reports and subsequently compare different companies in an industry remains lacking.

To enable automation, one approach that has emerged in prior research, albeit in adjacent fields (e.g., privacy), pertains to the creation of subject-specific dictionaries. Such dictionaries can form a basis for objectively evaluating the sustainability reporting practices of an organization. For instance, Vasalou et al. [30] developed a privacy dictionary to analyze research in social science, technology design, and policymaking. Their dictionary was created from automated content analysis of privacy-related texts. The data set for the research was collected through (a) interviews and focus groups' patterns of privacy, and (b) self-reported privacy concerns. Then the data set was processed by category-frequency software to develop a privacy dictionary. On the sustainability front, Deng et al. [31] built a dictionary specific to Green IT practices, by applying semi-automated methods similar to our approach. The dictionary was based on a corpus of 38 documents containing text gleaned from the environmental section of corporate sustainability reports of 49 Fortune 500 IT companies identified in the 2015 list. The researchers created a dictionary of 302 words that were manually pruned by coders and assigned the keywords to a set of 11 categories. One may note that the literature currently has only a few subject-specific dictionaries. In addition, the generation of semi-automated dictionaries through text analytics appears to be rare in general for analyzing corporate reporting practices.

### 2.3. Information Entropy as a Measure of Transparency

Entropy is defined as the measure of uncertainty of a random variable [32]. Previous research has used entropy to forecast the transparency of companies regarding information disclosure [33]. Information-entropy has also been used to analyze languages like Chinese [34] and Japanese [35] and for other linguistic topics [36]. This research builds upon concepts of information entropy to establish a measure of transparency of corporate reporting. More specifically the Kullback–Leibler divergence measure [32] is used to assess corporate transparency. The Kullback–Leibler divergence of a random variable is a measure of the difference between the two probability distributions, one of which is the "true" distribution, and the other is the distribution being compared [32]. Therefore, in a document space of information, entropy unifies the relative conclusion about uncertainty measures in complete and incomplete information systems [37]. The use of entropy to measure corporate transparency, especially in regard to sustainability reporting, does not appear to have been reported in the extant literature.

To summarize, the literature review has shown that some subject-specific dictionaries have been developed in the past, some of which have also used automated approaches. However, apparently except for the Green IT dictionary [31] no dictionary has been built through text analytics for the field of sustainability reporting. In addition, the application of automated dictionaries to contexts similar to the measurement of corporate transparency has not been reported. To measure corporate transparency, researchers have taken different approaches, but they are mostly based on subjective measures focused on business and operational strategies [38]. While the concept of entropy and its use in measuring transparency in linguistics and other fields was explored, an important finding is the lack

of application of information entropy to measure transparency in corporate reporting, especially in the domain of sustainability. The aforementioned gaps in the extant literature therefore motivated and guided the research methodology and objectives.

## 3. Research Methodology

This section discusses the underlying research methodology to find answers to the two research questions stated in the introduction. It consists of three phases: (i) Build and validate a dictionary of subject-specific keywords (in this case sustainability); (ii) develop a related information theoretical transparency measure for corporate reports; and (iii) demonstrate an application of the measure to study sustainability-related behavior of firms in the energy industry.

Building a sustainability-specific dictionary in a systematic way involves compilation of a raw corpus of source documents such as seminal subject-specific articles, authoritative subject-specific books, social media outlets and websites of NGOs, and sustainability-specific reports of selected companies. The application of data filtering and keyword extraction methods yields a dictionary of most frequent keywords, which are then further refined through expert input. The quality of both the source corpus and the dictionary of keywords is evaluated through multiple methods including statistical testing and external evaluation by subject experts. The probability distribution of terms in the sustainability dictionary serves as a basis for developing an entropy-based metric of corporate transparency in sustainability reporting. In this section, a theoretical definition of such a metric is presented followed by a brief discussion of its application to study transparency of corporate reports of firms in the energy sector.

### 3.1. Building and Validating a Sustainability Dictionary

Building and validating a sustainability dictionary involves the following steps: (i) The compilation of a corpus of documents that are rich in sustainability terms; (ii) removing common English terms and generating the most frequent words; (iii) refining the words through expert input and data quality checks; and (iv) determining the frequency count of the selected words and the frequency of all other non-selected words taken together in each document of the source corpus. The output of this phase is a dictionary probability space which serves as the reference for computing entropies as presented later in this section.

#### 3.1.1. Data Collection, Filtering, and Extraction

In order to develop the sustainability dictionary to analyze transparency in corporate reports, data were collected from five main sources: (i) 30 well-cited seminal papers in the topic of sustainability; (ii) ten relevant books on the topic of sustainability; (iii) a selection of the prior five years of sustainability reports of companies ranked by Forbes as most sustainable; (iv) content extracted from a selection of the thirty most influential non-Governmental organizations (NGO) that play a key role in promoting sustainability practices; and (v) the most recent 1000 tweets of each of these NGOs. Details of the sources of this data are available in Table A1 of Appendix A. Table 2 summarizes the results of data collection from the five sources: seminal papers, books, sustainability reports, NGO websites, and NGO tweets. A total of 140 text documents form the raw source corpus. The combined size of the document space exceeds 5 million words that yielded a set of 5636 most frequent initial words for further analysis.

**Table 2.** Descriptive statistics of corpus data.

| Corpus Data | Count |
|---|---|
| Number of source sets | 5 |
| Number of sustainability rich documents across all sources | 140 |
| Total corpus size (words) | 5,425,178 |
| Potentially relevant count of single words | 5636 |

Provalis text analytics software (https://provalisresearch.com) was used to process all the reports with a specific focus on removing common English words. Analysis yielded sets of unigrams (single words), bigrams (two words), and trigrams (three words). For this research, only unigrams (single words) were considered further. Mixing unigrams, bigrams, and trigrams in entropy calculations to derive a single measure of transparency poses numerous methodological challenges and therefore will be a subject of future research. The words with a frequency count of greater than 10 from the entire corpus were merged to yield a list of 5636 words. This list was further inspected manually with the help of a sustainability subject expert to remove insignificant and erroneous words. This reduced the list to 3905 words. Their frequency of occurrence in the merged corpus was used to rank the words, and the word list was further trimmed by applying the Pareto principle [39], which states that 80% of the effects come from 20% of the causes. In this case, the 80% cutoff resulted in 778 words.

Three criteria have been advanced by text analytics researchers to evaluate the goodness of a source corpus for building a dictionary of keywords—relevant, appropriate, and complete [31]. To be relevant the corpus must be consistent with the theme of the dictionary. Since corporate sustainability reports may encompass a variety of industries, this study made sure that the source corpus spans a wide variety of seminal documents such as journal papers, books etc. Deng et al. [31] built a dictionary specifically for Green IT which was manually verified by coders. The resulting dictionary is half the size of our final dictionary because of its specialized nature. Given that textual analysis of corporate sustainability reports is our ultimate goal, our dictionary is appropriate subject to the limitations mentioned later in this paper. We tested the completeness of our dictionary by comparing the frequency distribution of words by adding more source documents from each category. We observed that the frequency distribution of words did not change in a statistically significant manner, hence suggesting that our dictionary is complete relative to our corpus selection criteria. In the results section, we also discuss how the quality of the corpus was further evaluated resulting in dropping tweets as source data.

### 3.1.2. Quality Evaluation

The source corpus documents were randomly partitioned into two subsets and the aggregated frequency counts of the selected dictionary words in each subset were determined. A chi-square test was used to compare the two samples. The hypothesis was that the two samples are not independent. The resulting *p*-value is less than 0.001, so the related null hypothesis that the samples are independent can be rejected.

In order to further ascertain the veracity and sustainability relevance of the candidate unigrams statistically identified, the unigrams were independently evaluated by three experts. The invited experts were prominent scholars in sustainability, with an established track record in research and publications. To accommodate for breadth and diversity, the experts selected held positions in America, Asia, and Europe, and had their primary research in economic/managerial, environmental, and social dimensions of sustainability, respectively, along with multidimensional and multidisciplinary sustainability research. Each of the experts was sent a list of the top 778 unigrams. The list was sorted alphabetically so as to eliminate any potential bias in rendering their evaluation. The experts were advised of the scope of the research and asked to rate each word based on a 10-point Likert scale where 0 is not relevant to sustainability and 10 is highly relevant to sustainability. The experts were compensated for providing their evaluations. Once all the ratings were received, the statistical significance of each word was calculated. Words that the three experts considered not or least relevant were removed, resulting in a final dictionary of 665 sustainability relevant unigrams.

### 3.1.3. A Dictionary Probability Space and Information Entropy Measure

**Definition 1.** *Let U be the set of all sustainability-related documents (including corporate reports) containing words from the English language and R be a set of all words (unigrams) from the English language applicable to this set U.*

**Definition 2.** *Let S⊂ R be a dictionary of m−1 words generated by filtering out sustainability-specific keywords from R and $w_{i = 1 \dots m-1} \in S$ represent each final chosen dictionary word.*

The results of filtering the corpus is a set of keywords in *S* that form our sustainability dictionary. This dictionary probability space relative to the source corpus can be defined as follows.

**Definition 3.** *Let C ⊂ U be the documents that form the source corpus for deriving our dictionary and C = {C_1, C_2, . . . , C_N}.*

The probability distribution of words in each corpus document $C_{j = 1 \dots N}$ is derived as follows. First, the probability of occurrence of each word $w_i \in S$ in $C_j$ is obtained by dividing the frequency count of that word in the document by the total count of words in that document. This yields $m − 1$ probabilities. The *m*th probability term is then calculated as the ratio of the count of the remaining words to the total word count. This is a marginal probability representing all the remaining words in $C_j$ that are not in *S*. This is formally stated as follows:

**Definition 4.** *Let $c_j = (c_{j1}, \dots , c_{jm})$ be a probability distribution defined over R representing the distribution of words in a document $C_{j = 1 \dots N} \in C$ in the corpus. Let $c_{j1}, \dots , c_{jm-1}$ represent the probabilities corresponding to the dictionary words $w_{i = 1 \dots m} \in S$ and $c_{jm}$ represent the marginal probability of the rest of the words in the document $C_j$.*

The dictionary probability space comprising of these N probability distributions $c_{j = 1 \dots N}$ across the corpus documents $C_{j = 1 \dots N}$ form our reference for computing an entropy measure of transparency. These probability distributions serve as a basis for evaluating the quality of the corpus and the quality of our measure of transparency.

**Definition 5.** *Let $q = (q_1, \dots , q_m)$ be a probability distribution defined over R, such that $q_1, \dots , q_{m-1}$ represents the discrete probability of the m − 1 words in S, and $q_m$ represents a marginal probability of words in (R-S). Each probability $q_i$ represents the aggregated frequency of a word $w_i$ in C, the entire corpus set.*

**Definition 6.** *The Shannon entropy [32] of a random variable associated with q is defined as:*

$$H(q) = - \sum_{i=1}^{i=m} q_i * log(q_i) \tag{1}$$

Shannon entropy is a measure of the uncertainty associated with the outcome of an experiment, the possible results of which follow the distribution *q*. If log is computed to the base 2, entropy is measured in bits and if the natural logarithm is used, entropy is expressed in nats. In our case, the occurrence of the dictionary words constitutes the experiment outcome, and Shannon entropy measures the inherent uncertainty associated with that occurrence of the selected dictionary words in a document. The natural logarithm was used for our computations. Our interest lies in further development of this concept into a divergence measure associated with the probability distribution of words in a corporate document and the probability distributions $c_{j = 1 \dots N}$.

**Definition 7.** *Let p = ($p_1$, ... , $p_m$ ) be another probability distribution defined over R representing the distribution of words in a corporate report. Let $p_1$ ... $p_{m-1}$ represent the probabilities of those dictionary words in S that are in the report and $p_m$ represent the marginal probability associated with words in (R-S)*

*The Kullback–Leibler divergence from the probability distribution q to p is defined as follows [32]:*

$$D(p||q) = - \sum_{j=1}^{j=m} p_j * log(q_j/p_j) \qquad (2)$$

*Assumptions:*

1. *$q_j > 0$; i.e., the words in **S** have a nonzero probability of occurrence in **R.***
2. *$0 \, log \frac{0}{0} = 0$*
3. *$p \, log \frac{p}{0} = \infty$*

It is important to note that $D(p||q)$ is not equal to $D(q||p)$. Further $D(p||q) > =0$ [32].

**Definition 8.** *A measure of transparency T of a corporate report is defined on R as the Kullback–Leibler divergence (also called relative entropy) from the probability distribution q associated with our reference dictionary (see Definition 5) to the corresponding distribution p associated with the corporate report (See Definition 7).*

$$T = D(p||q) = - \sum_{j=1}^{j=m} p_j * log(q_j/p_j) \qquad (3)$$

*3.2. Interpreting and Applying the Entropy Measure in Practice*

If organizations are very transparent about their sustainability, the expectation is that their corporate reports are rich in sustainability-specific keywords, i.e., probability of occurrence of dictionary keywords *p* is close, in an information-theoretical sense, to the dictionary probability *q*. A measure of this distance between the two probability distributions is referred to as D(*p*||*q*), the divergence from *q* to *p*. This may also be characterized as the information gain achieved if q would be used instead of p. The use of KL divergence, in the context of machine learning is well accepted as more dynamic and robust than the use of Gini Index or Bayesian diagnosticity [40,41]. It is the only divergence that belongs to multiple classes of statistical divergences satisfying certain unique properties that make it widely used in statistical and machine learning applications [42].

Grounded in information theory, one may argue that *T* serves as a measure of how transparent corporations are in their sustainability reports considering the probability distribution of words in S, the sustainability dictionary, as the base reference. The greater (lower) the value of *T* the lower (higher) the transparency. Our definition of *T* considered entropy computation across the aggregated probability of each dictionary word across the entire corpus documents. This relies on the assumption that the source corpus documents were independently generated which is true in a broad sense. There also arises the need to capture information uncertainties across the source corpus. In order to evaluate the quality of the measure in relation to the corpus one may also rely on this independence assumption and compute the measure individually across each source document in the corpus and use the average measure.

Now that the definitions are declared, the next section will detail their application.

## 4. Results

*4.1. Entropy-Based Evaluation of the Sustainability Dictionary*

After building the sustainability dictionary by analyzing words from five different document source sets, the quality of the original corpus documents was studied. Recall that the probability distribution $c_j = (c_{j1}, ... , c_{jm})$ for each corpus document $C_j$ was derived by dividing the frequency count of the corresponding word $w_i$ in a document by the total count of words in that document, which yielded $m-1$ probabilities. The $m^{th}$ probability term was then calculated as the ratio of the count of the remaining words to the total word

count. To evaluate the quality of the source corpus, the Shannon entropy associated with each $c_j$ was calculated, and ANOVA tests performed to test the difference in mean entropies of the five source corpus data sets to determine their relative quality. The mean entropies for the source documents in each of the five sources are plotted in Figure 1a. The related one-way unpaired ANOVA tests for difference in means across the five different source groups shown in Figure 1b suggest the lack of a significant difference (*p*-value < 0.0001). Pairwise ANOVA tests results, however, showed that the mean entropy of tweet data documents exhibits significant differences with both seminal papers and books while others appeared to be pair-wise comparable. Given that each tweet document was generated from 1000 tweets, it is plausible that tweets are not rich in sustainability content and therefore do not constitute a rich source of data for building the sustainability dictionary. This is likely due to the intrinsic nature of microblogging which limits the textual context for encoding messages, hence leading to compromises in communication. This prompted the removal of the tweet data probability distribution from the dictionary probability space for the computation of the measure of transparency.

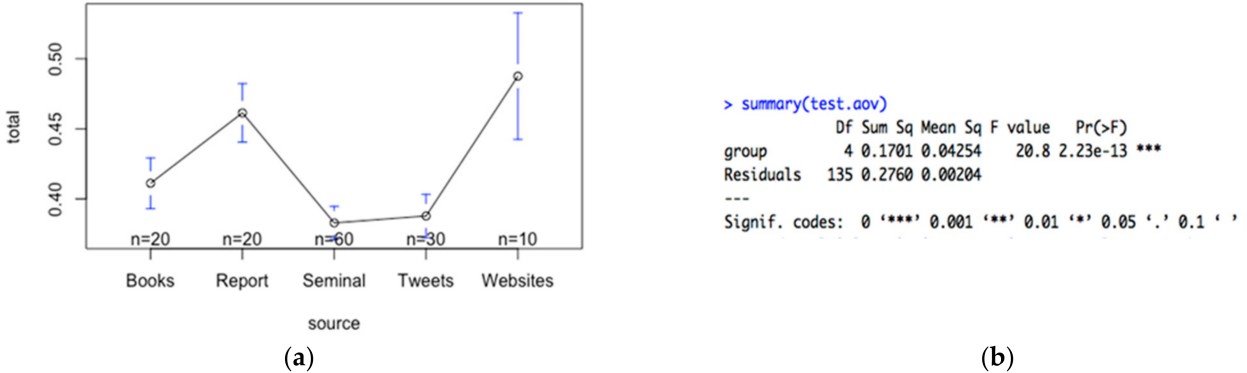

**Figure 1.** (**a**) Mean plot for entropy distribution of corpus documents; (**b**) ANOVA test results for entropy distribution of corpus documents.

### 4.2. Applying the Measure of Transparency

#### 4.2.1. Transparency Measures for Selected OPEC and Non-OPEC Firms

In order to illustrate the effectiveness of the transparency measure, the annual sustainability reports of the top 15 OPEC and non-OPEC energy companies (by revenue) were compiled. If any of the top 15 companies had gaps in their annual report series, the company was replaced with the next company in the list as sorted by revenue. This resulted in 4 OPEC companies and 2 non-OPEC companies being replaced. Table 3 shows all the companies that were included in the final list for comparative purposes. Reports published in the years 2015–2017 were used for the research. An additional point to note is that significant restructuring happened in the energy industry (for instance the constitution of OPEC and non-OPEC groups) post 2017. This also significantly impacts the homogeneity of the dataset if we were to include more recent reports in this study.

The sustainability reports for each company were then analyzed to determine the associated probability distribution p (see Definition 7). The transparency measure for each report was computed based on the probability distribution q (see Definition 5) and the formula for the Kullback–Leibler divergence (see Definition 8). The resulting dataset was further analyzed by using ANOVA methods to test for differences in means as described in the next subsection. The results of the one-tailed tests indicated a significant difference in the transparency measure for non-OPEC producers in comparison with that for OPEC producers.

**Table 3.** List of the top 15 OPEC and non-OPEC energy companies by revenue.

| OPEC Companies | Non-OPEC Companies |
| --- | --- |
| Saudi Aramco | Sinopec |
| Kuwait Petroleum Corporation | Rosneft |
| PDVSA | PetroChina |
| National Iran Oil Company | Exxon Mobil |
| Sabic | Royal Dutch Shell |
| Ras Gas | BP |
| QP | Total SA |
| ADNOC | Lukoil |
| Sonangol | Eni |
| Arabian Gulf Oil Company | Gazprom |
| Nigerian National Petroleum Company | Valero Energy |
| Aiteo | Petrobras |
| Emirates National Oil Company | Chevron Corporation |
| Qatar Gas | PEMEX |
| Kuwait Energy | Petronas |

4.2.2. Examining the Quality of the Transparency Measure

In order to evaluate the goodness of the information-theoretical measure, it was deemed useful to compute the transparency score for two documents that are likely to be boundary cases for sustainability-rich keywords. Two such documents were chosen: (i) Moby Dick (https://www.gutenberg.org/files/2701/2701-h/2701-h.htm); and (ii) a web-scraped compilation of all Wikipedia pages on the topic of sustainability (https://en.wikipedia.org/wiki/Index_of_sustainability_articles). The assumption is that Moby Dick, being a work of fiction not purposefully created with sustainability in mind must have a relatively low transparency, while the Wikipedia pages taken together, being rich in sustainability-specific keywords must be relatively high in transparency. The computations yielded values of $T_{\text{Moby Dick}} = 0.2238$ and $T_{\text{Sustainability Wiki}} = 0.14132$. A one-tailed t-test confirmed that these values are indeed significantly different in terms of the dictionary probability space, specifically the value of $T$ for Moby Dick is significantly greater ($p$-value $< 2.2 \times 10^{-16}$) than the value of $T$ for the sustainability Wiki pages document. This provides initial evidence about the inherent quality of the transparency measure, specifically that it sufficiently discriminates between two documents that are extreme case examples.

4.2.3. Comparing OPEC with Non-OPEC

Computing the transparency measure for all the companies in the OPEC and non-OPEC list, as detailed earlier, facilitates a comparative analysis of their sustainability reporting behavior. It followed a four-step process.

Step1—Test for normality of underlying data: Normality of the data is one assumption for the continued analysis using one-way ANOVA methods. A box plot of the values of $T$ for all companies in each group revealed the presence of a total of 6 outliers across all 90 reports in the dataset. These were removed for our further analysis. The resulting probability distributions (with outliers removed) of both OPEC and non-OPEC organizations shown in Figure 2 suggest that both are normal distributions.

The corresponding Q-Q plots for both datasets are very close to the normal line and within the confidence bands. A related Shapiro–Wilk test for normality yields high $p$-values of 0.4348 and 0.2627 respectively for OPEC and non-OPEC. The related null hypothesis that the transparency scores are normally distributed for both groups cannot be therefore rejected. Given that the reports were independently generated it is plausible that within the groups they follow a normal distribution (except for the few outliers). Our main hypothesis is that these two datasets that characterize the transparency behavior are different. An examination of the related descriptive statistics sheds further light on this.

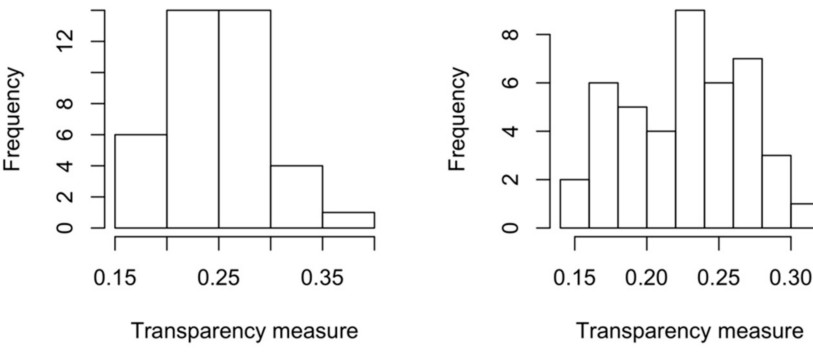

**Figure 2.** Histograms of transparency score—PEC vs. non-OPEC.

Step 2—Compute the descriptive statistics: The overall mean transparency score of 0.2338 for non-OPEC organizations is less than the corresponding score of 0.3031 of OPEC companies. The median transparency scores (OPEC = 0.2608, Non-OPEC = 0.2319), the minimum transparency scores (OPEC = 0.1694, Non-OPEC = 0.1487), and the maximum transparency scores (OPEC = 0.8722, Non-OPEC = 0.3925) all indicate clear difference between the two groups.

Step 3—Perform the one-way ANOVA test: A one-way ANOVA test to compare the means helps further validate the difference. The alternative hypothesis was that the mean transparency score for non-OPEC companies is different than that of OPEC companies. The related test results are shown in Figure 3. The null hypothesis that the means are not different can be rejected. Further, a corresponding Welch's one-tailed two sample t-test (*p*-value = 0.007039) shows that the mean transparency measure for OPEC companies is significantly greater than that of the non-OPEC companies.

## Analysis of Variance Table
### Response: Value

|        | Df  | Sum Sq    | Mean Sq   | F value | Pr( > F)  |
|--------|-----|-----------|-----------|---------|-----------|
| Var    | 1   | 0.011733  | 0.0117330 | 6.3857  | 0.01348 * |
| Residuals | 80 | 0.146991 | 0.0018374 |         |           |

Signif. codes:   0 '***' 0.001 '**' 0.01 '*' 0.05 '.' 0.1 ' ' 1

**Figure 3.** ANOVA test to compare the means of OPEC and non-OPEC companies

Step 4—Perform pairwise ANOVA tests to study the trend: The trend of corporate transparency in sustainability reporting was further examined for the three years for which corporate reports of the 15 selected companies were selected for our study. As mentioned earlier, the data for this research were collected in the summer of 2018. A boxplot of the trend for the prior three years for each group is shown in Figure 4 indicating increasing mean transparency measure over the years. However, a within groups one-way ANOVA across the three years do not suggest any significant change in the mean transparency measure for both OPEC and non-OPEC indicating that transparency behavior is consistent over the years. The result of a pairwise ANOVA across the three years also indicated this consistency in behavior.

These results clearly point to a significant and consistent difference in transparency behavior between OPEC and non-OPEC businesses in terms of corporate sustainability reporting over the three-year period of our study.

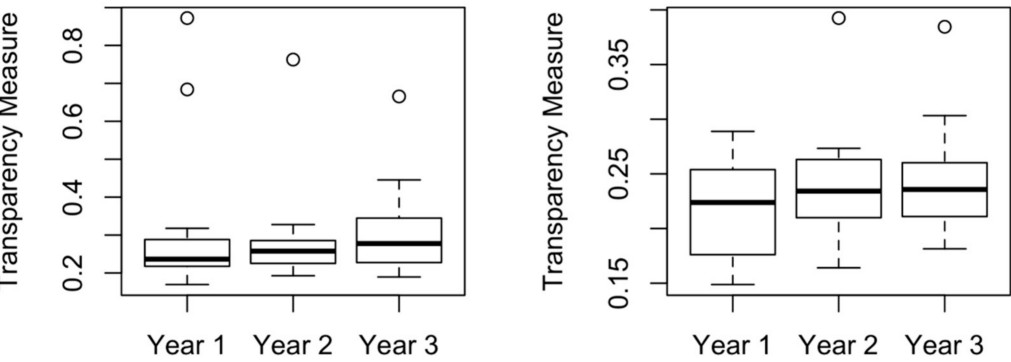

**Figure 4.** Box plot of transparency measure (OPEC vs. non-OPEC).

## 5. Discussion

Based on the established importance of transparency for achieving sustainability, a principle entrenched in international treaties (e.g., Paris Agreement 2015) and corporate best practices, this study aimed to address two key research questions: (1) What is an effective measure of corporate transparency in sustainability reporting? and (2) How can this measure be applied to rate and compare the organizations in a specific industry sector?

### 5.1. Addressing the Subjective Nature of Existing Measures

In addressing the first research question, the study reviewed the current literature on measures pertaining to sustainability. The extant literature reveled several measures, yet with substantive limitations. While a majority of these measures encompassed several dimensions [14,16,19], they had two key limitations restricting their effectiveness in the digital economy. The first limitation relates to their scope, with none of the current measures effectively capturing the broad holistic scope of sustainability, which should encompass economic, environmental, and social dimensions. Therefore, while informative, these measures were narrow and ineffective in capturing the multidimensional essence of sustainability. The second limitation relates to the subjectivity in applying such measures. This subjectivity necessitates the review and interpretation of a human actor in order to render a judgment on the degree of sustainability. As noted above, the proliferation of casual commentators and pseudo-experts, and the time-consuming nature required to provide a thorough analysis of a multitude of reports, have both resulted in complexities that deter from objectivity and timeliness. Consequently, this study, inspired by research in adjacent fields [30], responded to the first research question, by developing a rigorous measure, manifested in a sustainability dictionary. The dictionary was derived through sustainability-rich sources to which statistical methods were applied and unigrams identified. The unigrams were then statistically confirmed and further validated by experts in the field of sustainability.

### 5.2. Applying the Objective Measure to Compare Producers in the Energy Sector

The second research question took an empirical perspective in considering the operationalization of the sustainability dictionary measure developed in the study. A similar recent example of empirical analysis of reporting is presented by Dumitru et al. [43] who derived 37 terms from the European Directive (2014/95/EU) on disclosure by energy companies. They found a modest increase in the use of keywords relevant to the European Directive in the companies' annual reports. This study applies a similar intent, yet with a validated dictionary derived from multiple influential sources and substantive statistical models based on information entropy. Specifically, our study adopted Kullback–Leibler divergence to objectively assess the sustainability reports of key fossil-fuel producers in the energy sector. This process was automated to ensure objectivity and timeliness—both essential criteria for transparency in sustainability reporting in the digital economy. To illustrate the efficacy of this approach, a comparative analysis of OPEC vs. non-OPEC major

producers was undertaken. The outcome of the comparative analysis demonstrates higher transparency among non-OPEC producers relative to OPEC producers. This could point to normative pressures for transparency applied by regulators, the media, NGOs, and/or the community at large on such producers to report on their sustainability operations. It seems that cartels are immune to such pressures because of their size, centralized decision-making processes, and combined political influence.

*5.3. Comparison with CSR Rankings*

To further examine these findings, the authors collected data pertaining to corporate social responsibility (CSR) rankings of the companies within the scope of the study from CSRHub, which reports to provide "Transparent ratings and rankings of 17,660 companies from 145 countries, driven by 724 industry-leading CSR/ESG data sources including ESG analyst, crowd, government, publication, and not-for-profit data" (https://www.csrhub.com/). From the available CSRHub data collected, the following were observed: (1) Only three companies in the OPEC group received a full CSR rating (OPEC full CSR rating-3/15), others received either partial rating (OPEC partial CSR rating-4/15) or no rating at all (OPEC no CSR rating-8/15). Conversely, only two of the non-OPEC companies were not rated (non-OPEC full CSR rating-13/15). This gives credence to the point raised above in relation to normative pressures. Even with the volume of data and the diversity of data sources, most OPEC data on sustainability are missing from CSRHub, while most non-OPEC data are available.

Importantly, this study bypasses such limitation. Most OPEC and non-OPEC producers do publish sustainability reports, which could provide empirical data for a more thorough analysis and assessment of transparency in sustainability, beyond the simplistic binary perspective of available/non-available public scores. The approach taken in this study transcends this by objectively and automatically assessing the degree of transparency in sustainability reporting, which enables a richer analysis and more meaningful results that are better aligned with sustainability in the digital economy.

## 6. Research Contributions

*6.1. Theoretical Contributions*

The study holds important theoretical contributions. To the authors' knowledge, this is the first study to develop a sustainability dictionary and validate such dictionary with experts. The dictionary development was in itself derived from a range of sustainability rich sources, adding to the relevance of the dictionary. While the dictionary can be applied in multiple contexts to assess sustainability, e.g., in a range of industries or organizations including CSR, a core theoretical contribution emanates from the method by which such dictionary was developed. This points to a methodological contribution that could be adopted to develop dictionaries in other similar contexts such as e-healthcare, consumer online behavior, and e-governance. Moreover, an empirical contribution is manifested through the results on OPEC/non-OPEC producers which, while based on historical data for illustrative purposes, points to interesting dynamics between institutional powers and market forces when it comes to transparency in sustainability reporting.

*6.2. Practical Contributions*

From a practical standpoint, the automated measure prototype developed in this research, can, upon further development and diffusion, be used to generate timely and objective assessments. This could prove particularly valuable in multiple contexts and for a range of stakeholders in the digital economy, including investors, the media, NGOs, governments and regulators, and the public at large. Agudelo et al. [44] point to multiple drivers motivating energy companies to engage in responsible behavior, including internal environmental commitments, organizational, strategy and culture, and risk prevention and management. This therefore identifies the intra-organizational actors as equally important stakeholders in sustainability reporting transparency. Consequently, the authors contend

that the diffusion of such tool to assess transparency in sustainability reporting could itself be transformative, to both intra-organizational and extra-organizational stakeholders in the energy sector, as it could make it easy to assess and compare the corporate performance, leading to internal, public, and regulatory pressures to improve corporate performance and drive it toward sustainability, over time.

## 7. Conclusions, Limitations, and Future Research

Transparency is fundamental to corporate sustainability. The aim of this research was to develop deeper understanding of two core aspects—developing an appropriate measure of corporate transparency in sustainability reporting and applying it to examine the sustainability reporting practices of the energy sector.

First, the research demonstrated the development of a sustainability dictionary from rich sources of sustainability data. This extends current approaches reported in Table 1, which focused on factors and indicators, by providing a novel approach in creating an objective benchmark. The sustainability dictionary was built by using text analytics and validated through split-sampling and related chi-square tests. The probability distribution associated with the sustainability dictionary words was computed based on their frequency of occurrence in the source documents, and the final choice of words for the dictionary was made with the validation of multiple subject matter experts. It served as the basis for deriving an information-theoretical measure of corporate transparency.

Second, this was then applied for illustrative purposes to compare transparency in sustainability reporting in the energy sector. This builds on previous studies (e.g., [15,17]) by providing an information entropy driven automation. This helped validate the measure and to gain insights into the potential normative pressures on non-OPEC producers to be more transparent in their reporting.

A major limitation of this research is that the dictionary is based on unigrams (single words). Although digrams and trigrams were filtered out of the source data, they were not used in developing the transparency measure. In addition, the research only considered the absolute frequency of the occurrence of keywords, not dispersion or concurrence. Finally, the research assumed each data source was independent of the other sources. So, the influence of one data source over another data source in the sustainability corpus was assumed insignificant, which allowed the computation of mean entropies. These limitations can be addressed through future research that builds on the current findings and contributions. Specifically, future work could explore the use of Markov models to capture n-grams relevant to the domain. Through using digrams and trigrams, the transparency score can be tuned further. Also, future work can include dispersion of words and the co-occurrence probabilities of each word. Future research can also include investigating other forms of entropy—e.g., Tsallis entropy or other forms of divergence measures to score corporate reports. Moreover, applying the current methodology longitudinally to future reports in the energy sector would assist in tracking transparency progress over time, hence uncovering emergent dynamics in sustainability reporting.

Through the use of the derived sustainability dictionary and the entropy-driven methodology, different industries can be compared to determine their relative transparency in sustainability reporting. Also, a future longitudinal study comparing OPEC with non-OPEC will also be interesting. Higgins el al. [45] contend that sustainability reports are largely a legitimacy-seeking façade, yet with potential. The contributions of this study amplify such potential, as they provide the means for thorough, automated, and rapid analysis of sustainability reports, which could provide a conduit for transformation. The research can also be extended to other areas such as privacy, security, and corporate accounting practices. Building upon the findings it would be essential to progress in developing mechanisms for transparency as it would produce objective and timely assessments that would drive sustainability throughout the digital economy.

**Author Contributions:** All the authors had an equal contribution to all parts of the manuscript. All authors have read and agreed to the published version of the manuscript.

**Funding:** This work was partly made possible by two seed research grants from the Qatar Foundation to two of the authors. The statements made herein are solely the responsibility of the authors.

**Data Availability Statement:** Appendix A provides details of where data supporting reported results can be found.

**Acknowledgments:** The authors wish to acknowledge the support provided by Devi Kurup in her role as research assistant.

**Conflicts of Interest:** The authors declare no conflict of interest.

## Appendix A

**Table A1.** Core content for the sustainability dictionary corpus.

| Source Set 1: Research Publications on Sustainability | |
|---|---|
| Source | Citation Count |
| Malhotra, A.; Melville, N.P.; Watson, R.T. Spurring impactful research on information systems for environmental sustainability. *MIS Q.* **2013**, *37*, 1265–1274. | 259 |
| Martins, C.I.M.; Eding, E.H.; Verdegem, M.C.; Heinsbroek, L.T.; Schneider, O.; Blancheton, J.P.; Verreth, J.A.J. New developments in recirculating aquaculture systems in Europe: A perspective on environmental sustainability. *Aquac. Eng.* **2010**, *43*, 83–93. | 694 |
| Mol, A.P.J. Boundless biofuels? Between environmental sustainability and vulnerability. *Sociol. Rralis* **2007**, *47*, 297–315. | 238 |
| Chappells, H.; Shove, E. Debating the future of comfort: environmental sustainability, energy consumption and the indoor environment. *Build. Res. Inf.* **2005**, *33*, 32–40. | 486 |
| Basiago, A.D. "Economic, social, and environmental sustainability in development theory and urban planning practice." *Environ.* **1998**, *19*, 145–161. | 491 |
| Barbier, E.B.; Markandya, A.; Pearce, D.W. Environmental sustainability and cost-benefit analysis. *Environ. Plan. A* **1990**, *22*, 1259–1266. | 272 |
| Morelli, John. Environmental sustainability: A definition for environmental professionals. *J. Environ. Sustain.* **2011**, *1*, 2. | 609 |
| Goodland, R.; Daly, H. Environmental sustainability: Universal and non-negotiable. *Ecol. Appl.* **1996**, *6*, 1002–1017. | 798 |
| Moldan, B.; Janoušková, S; Hák, T. How to understand and measure environmental sustainability: Indicators and targets. *Ecol. Indic.* **2012**, *17*, 4–13. | 882 |
| McBride, A.C; Dale V.H.; Baskaran L.M.; Downing M.E.; Eaton L.M.; Efroymson R.A.; Garten Jr. C.T.; Kline K.L.; Jager H.I.; Mulholland P.J.; Parish E.S. Indicators to support environmental sustainability of bioenergy systems. *Ecol. Indic.* **2011**, *11*, 1277–1289. | 228 |
| Melville, N.P. Information systems innovation for environmental sustainability. *MIS Q.* **2010**, *34*, 1–21. | 1250 |
| Sarkis, J. Manufacturing's role in corporate environmental sustainability—Concerns for the new millennium. *Int. J. Oper. Prod. Manag.* **2001**, doi:10.1108/01443570110390390. | 531 |
| Kern, F.; Smith, A. Restructuring energy systems for sustainability? Energy transition policy in the Netherlands. *Energy Policy* **2009**, *36* 4093–4103. | 580 |
| Elliot, S. Transdisciplinary perspectives on environmental sustainability: a resource base and framework for IT-enabled business transformation. *MIS Q.*, **2011**, *35*, 197–236. | 483 |
| Orlitzky, M.; Siegel, D.S.; Waldman D.A. Strategic corporate social responsibility and environmental sustainability. *Bus. Soc.* **2012**, *50*, 6–27. | 731 |
| Wognum, P.N.; Bremmers, H.; Trienekens, J.H.; van der Vorst, J.G.; Bloemhof, J.M. Systems for sustainability and transparency of food supply chains–Current status and challenges. *Adv. Eng. Inform.* **2011**, *25*, 65–76. | 350 |
| Dangelico, R.M.; Pujari, D. Mainstreaming green product innovation: Why and how companies integrate environmental sustainability. *J. Bus. Ethics* **2010**, 95, 471–486. | 925 |
| Kemp, R. Technology and the transition to environmental sustainability: the problem of technological regime shifts. *Futures* **1994**, *26*, 1023–1046. | 911 |
| Goodland, R. The concept of environmental sustainability. *Annu. Rev. Ecol. Syst.* **1995**, *26*, 1–24. | 1832 |
| Jenkins, H.; Yakovleva, N. Corporate social responsibility in the mining industry: Exploring trends in social and environmental disclosure. *J. Clean. Prod.* **2006**, *14*, 271–284. | 1164 |
| Schaltegger, S.; Csutora, M. Carbon accounting for sustainability and management. Status quo and challenges. *J. Clean. Prod.* **2012**, *36*, 1–16. | 270 |
| Holling, C.S. Understanding the complexity of economic, ecological, and social systems. *Ecosyst.* **2001**, *4*, 390–405. | 4556 |
| Ostrom, E.; Burger, J.; Field, C.B.; Norgaard, R.B.; Policansky, D. "Revisiting the commons: local lessons, global challenges." *Sci.* **1999**, *284*, 278–282. | 3420 |
| Sharma, S.; Vredenburg, H. Proactive corporate environmental strategy and the development of competitively valuable organizational capabilities. *Strateg. Manag. J.* **1998**, *19*, 729–753. | 3006 |
| Grimm, N.B.; Faeth, S.H.; Golubiewski, N.E.; Redman, C.L.; Wu, J.; Bai, X.; Briggs, J.M. Global change and the ecology of cities. *Sci.* **2008**, *319*, 756–760. | 4842 |
| Milne, M.J.; Gray, W. W(h)ither ecology? The triple bottom line, the global reporting initiative, and corporate sustainability reporting. *J. Bus. Ethics* **2013**, 118, 13–29. | 872 |
| Stechemesser, K.; Guenther, E. Carbon accounting: a systematic literature review. *J. Clean. Prod.* **2012**, *36*, 17–38. | 242 |
| Cheng, B.; Ioannou, I.; Serafeim, G. Corporate social responsibility and access to finance. *Strateg. Manag. J.* **2014**, *35*, 1–23. | 1799 |
| Delmas, M.; Blass, V.D. Measuring corporate environmental performance: the trade-offs of sustainability ratings. *Bus. Strategy Environ.* **2010**, *19*, 245–260. | 366 |
| Frias-Aceituno, J.V.; Rodriguez-Ariza, L.; Garcia-Sanchez, I.M. The role of the board in the dissemination of integrated corporate social reporting. *Corp. Soc. Responsib. Environ. Manag.* **2013**, *20*, 219–233. | 489 |

**Table A1.** *Cont.*

| Source Set 2: Books on Sustainability |
|---|
| Picon, A. *Smart Cities: A Spatialised Intelligence*, 1st ed.; Wiley: New Jersey, United States, 2015. |
| Ioris, A.A.R. *Agriculture, Environment and Development: International Perspectives on Water, Land and Politics*. Palgrave Macmillan: London, United Kingdom, 2016. |
| Eric G. Derouane.; Parmon, V.; Lemos, F.; Ribeiro, F.R. *Sustainable Strategies for the Upgrading of Natural Gas: Fundamentals, Challenges, and Opportunities*. Springer: New York, United States, 2005. |
| Sayler, G.S.; Sanseverino, J.; Davis, K.L. *Biotechnology in the Sustainable Environment*. Springer: New York, United States, 1997. |
| Posselt, G. *Towards Energy Transparent Factories*. Springer: New York, United States, 2016. |
| Heinrichs, H.; Martens, P.; Michelsen, G.; Wiek, A. *Sustainability Science: An Introduction*. Springer: New York, United States, 2016. |
| Mawhinney, M. *Sustainable Development: Understanding the Green Debates*. Blackwell Science: New Jersey, United States, 2002. |
| Dastbaz, M.; Strange, I.; Selkowitz, S. *Building Sustainable Futures: Design and the Built Environment*. Springer: New York, United States, 2016. |
| Kaushika, N.D.; Reddy, K.S.; Kaushik, K. *Sustainable Energy and the Environment: A Clean Technology Approach*. Springer: New York, United States, 2016. |
| Strange, T.; Bayley, A. *Sustainable Development: Linking Economy, Society, Environment*. OECD Publications: Paris, France, 2008. |

| Source Set 3: NGOs | |
|---|---|
| Name | Website |
| CERES | https://www.ceres.org/ (accessed 14 July, 2018) |
| HEIFER | https://www.heifer.org/ (accessed 14 July, 2018) |
| Forum for the Future | https://www.forumforthefuture.org/ (accessed 14 July, 2018) |
| Global Reporting | https://www.globalreporting.org/ (accessed 14 July, 2018) |
| CDP | https://www.cdp.net/en (accessed 14 July, 2018) |
| Natural Capitalism Solutions | https://natcapsolutions.org/ (accessed 14 July, 2018) |
| Rain Forest Action Network | https://www.ran.org/understory/ (accessed 14 July, 2018) |
| Greenpeace | https://www.greenpeace.org (accessed 14 July, 2018) |
| World Wildlife Fund | https://www.worldwildlife.org/ (accessed 14 July, 2018) |
| 350.org | https://350.org/ (accessed 14 July, 2018) |
| Note: Primary source for this data is the suggested list of the top 30 environmental NGOs—https://www.raptim.org/30-environmental-ngos-we-all-should-support (accessed 14 July, 2018) | |

| Source Set 4: Corporate Reports | |
|---|---|
| A selection of available reports in the CSR Hub (https://www.csrhub.com) for the period (2013–2017) from top 20 in Forbes List of the 100 most sustainable companies in the world. | Source: https://www.forbes.com/sites/jeffkauflin/2017/01/17/the-worlds-most-sustainable-companies-2017/?sh=417389604e9d (accessed 14 July, 2018) |

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
