# Peer review of "Objective Sustainability Assessment in the Digital Economy: An Information Entropy Measure of Transparency in Corporate Sustainability Reporting"

_sustainability, doi:10.3390/su13031054_

Round 1
Reviewer 1 Report
The paper investigate transparency of corporate sustainability reporting using an information entropy-based approach. The case study was conducted on companies in the energy sector. The paper is interesting, but some aspects need to be improved.
Literature review
I think the literature could be improved by adding recent research from the last two years. It should also be mentioned why entropy is better than other methods used to measure transparency with citation of other relevant papers.
There are many articles published in the last two years, some relevant articles can be found below, but you can search for more quotes.
Agudelo, M. A. L., Johannsdottir, L., & Davidsdottir, B. (2020). Drivers that motivate energy companies to be responsible. A systematic literature review of Corporate Social Responsibility in the energy sector. Journal of Cleaner Production, 247, 119094.
Dumitru, V. F., Jinga, G., Stănilă, O. G., & Dumitru, M. (2019, May). The impact of the European Directive 2014/95/EU on the energy companies’ disclosures. In Proceedings of the International Conference on Business Excellence (Vol. 13, No. 1, pp. 268-276). Sciendo.
Higgins, C., Tang, S., & Stubbs, W. (2020). On managing hypocrisy: The transparency of sustainability reports. Journal of Business Research, 114, 395-407.
Li, M., Wang, J., Li, Y., & Xu, Y. (2018). Evaluation of sustainability information disclosure based on entropy. Entropy, 20(9), 689.
Oncioiu, I., Petrescu, A. G., Bîlcan, F. R., Petrescu, M., Popescu, D. M., & Anghel, E. (2020). Corporate Sustainability Reporting and Financial Performance. Sustainability, 12(10), 4297.
Minor aspect - line 184 the citation (Cover & Thomas, 1991) must be numbered and after is need to check all numbering.
In my opinion figure 1 is not very suggestive, maybe you can find a better version.
3.1.1. Data collection - How were the five sources selected? Given that the data were collected two years ago, they are a bit old, it would be good to create an appendix for for the first four sources.
4.2.1 Transparency measures for selected OPEC and non-OPEC firms
Regarding the following sentence "Data was compiled in the summer of 2018 for the prior three years", is need to mention which are the three years: 2017, 2016 and 2015, or 2016, 2015 and 2014.
Discussions - The comparisons of the results obtained with other similar studies are missing you can try to find other research with similar results.
Given that the data were collected two years ago, it should be specified what is the relevance of the research results over time and whether the study would have been updated with another two years how much the research results have changed.
Author Response
Reviewer 1:
Comment 1:
I think the literature could be improved by adding recent research from the last two years.
Response:
We thank the reviewer for pointing out recent relevant studies. We found the studies informative. Besides these, we have added other references that are most recent and also one from the Sustainability journal (a total of 12 new references, 10 of which are recent). Several sections reflect these additions/changes - the literature review, discussion, contributions, and conclusion. These new references cover topics such as corporate information disclosure , the energy sector, analysis of corporate reports through content analysis and text mining, and justification of entropy as a measure primarily address reviewer comments.
Comment 2:
Minor aspect - line 184 the citation (Cover & Thomas, 1991) must be numbered and after is need to check all numbering.
Response:
This has been fixed.
Comment 3:
It should also be mentioned why entropy is better than other methods used to measure transparency with citation of other relevant papers.
Response:
We have now added the following statement along with 3 related references.
"The use of KL divergence, in the context of machine learning is well accepted as more dynamic and robust than the use of Gini Index or Bayesian diagnosticity (Burnham & Anderson, 2001; MacKay, 2003). It is the only divergence that belongs to multiple classes of statistical divergences satisfying certain unique properties that make it widely used in statistical and machine learning applications (Hobson, 1971). "
Comment 4:
In my opinion, figure 1 is not very suggestive, maybe you can find a better version.
Response:
Since Reviewer 3 also expressed the same opinion we have removed the figure and instead detailed the steps involved in the research in an itemized manner in the narrative providing better clarity.
Comment 5:
3.1.1. Data collection - How were the five sources selected? Given that the data were collected two years ago, they are a bit old, it would be good to create an appendix for the first four sources.
Response:
We thank the reviewer for this comment. Kindly note the following:
- Section 3.1.1 presents details of the five main sources for the corpus. We selected our corpus based on three criteria advanced by text analytics researchers - relevance, appropriateness, and completeness. This discussion has now been added.
- The completeness of the corpus was checked by doing a sensitivity analysis by adding more documents to the source corpus. This did not produce a statistically significant change (Chi-square test) in the word frequencies. The nature of the corpus and the objective of the dictionary to broadly encompass the subject of sustainability suggest that currency of the source corpus is unlikely to influence the composition of the dictionary.
- We asked our three subject experts to not only rank the word list but also comment on its adequacy by noting missing words. This did not yield any useful additional information suggesting that dictionary source is not heavily dependent on currency.
- Further, the paper also details how the quality of the corpus was evaluated using information entropy resulting in dropping tweets as a corpus source.
- We are currently working on a research publication based on other potential applications of the dictionary. An outcome of that effort will be the publication of the dictionary online along with detailed information of the source corpus.
Comment 6:
4.2.1 Transparency measures for selected OPEC and non-OPEC firms
Regarding the following sentence "Data was compiled in the summer of 2018 for the prior three years", is need to mention which are the three years: 2017, 2016 and 2015, or 2016, 2015 and 2014.
Response:
We have now added this information to the paper - Reports published in the years 2015-2017 were used for the research (Section 3.1.1).
Comment 7:
Discussions - The comparisons of the results obtained with other similar studies are missing you can try to find other research with similar results.
Response:
We thank the reviewer for pointing this out. Table 1 provides the initial baseline for our study, pointing out that existing approaches are dependent on expert input and measures are subjective. Our aim is to create an objective measure, the creation of which lends itself to automation.
Under results, Section 5.3 provides a comparison with CSRhub data to showcase the advantage of our approach and highlight the superiority of our findings. We have also added a contrast to the paper by Dumitru et al. in section 5.2. Given the novelty of our study, both in methodology and results, the pool of similar studies to directly compare with is limited. Section 5.1, 6.2, and 7 all provide contrasts, reflections, and extensions to prior research.
Comment 8:
Given that the data were collected two years ago, it should be specified what is the relevance of the research results over time and whether the study would have been updated with another two years how much the research results have changed.
Response:
This is a particularly insightful comment. We thank the reviewer. Our response to the reviewer's comment 5 addresses the currency of the dictionary. We now need to address the data related to OPEC vs non-OPEC. Indeed, applying a longitudinal perspective will assist in comparing not only across cartel/non-cartel dimensions but also temporally to see how the dynamics in the energy sector evolve over time. This will be particularly interesting, in order to discern how transparency in sustainability reporting is evolving. We have mentioned this as an opportunity for future research in section 7.
Additional point to note is that significant restructuring happened in the energy industry (for instance the constitution of OPEC and non-OPEC groups) post 2017. This also significantly impacts the homogeneity of the dataset if we were to include more recent reports.
Further, the case of the energy industry is used to showcase the potential of applying our proposed measure of transparency. The results validate two facets of our methodological approach: (1) the noted significant difference between the two groups has plausible explanation; (2) the comparison with the CSR rating also provides further validation of our methodology in terms of the inability of CSR rating to sufficiently differentiate the companies and the lack of CSR rating for most OPEC companies.
Overall, we sincerely thank all three reviewers for their insightful feedback. We have applied their constructive suggestions for improving the paper’s structure, argumentation, and clarity.
Sincerely,
The Authors
-------------------------------------------------------------------------------------
Reviewer 2 Report
The subject of the article is appropriate for the journal.
The abstract is correctly structured. OPEC must be specified the first time it appears. The methodology followed for said research must also be advanced.
The introduction is well structured and adequate.
In the theoretical review, the source of Table 1 must be specified. The number of sources consulted for the theoretical review should be expanded, reviewing relevant scientific articles.
In addition to the discussion, conclusions should be included, alluding to previous studies.
The format of the references should be reviewed.
Author Response
Reviewer 2:Comment 1:
The abstract is correctly structured.
Response:
We thank the reviewer for the comment.
Comment 2:
OPEC must be specified the first time it appears.
Response:
We have done this now starting in the abstract.
Comment 3:
The methodology followed for said research must also be advanced.
Response:
Based on Reviewer 1's feedback also on this, we have provided a a detailed overview of the methodology.
Comment 4:
The introduction is well structured and adequate.
Response:
We thank the reviewer for the comment.
Comment 5:
In the theoretical review, the source of Table 1 must be specified.
Response:
Table 1 was created by the authors and each study in the table is cited and included in the reference. We have added one more related reference to Table 1 to showcase the various types of measures of disclosure reported in the extant literature with the specific objective of pointing out that our aim is to develop an objective measure that lends itself to automation.
Comment 6:
The number of sources consulted for the theoretical review should be expanded, reviewing relevant scientific articles.
Response:
We thank the reviewer for this very useful comment. We have added three more references to justify the use of KL divergence as our theoretical measure. In addition, several new references have been added related to measures of disclosure in general under both the introduction and the literature review.
Comment 7:
In addition to the discussion, conclusions should be included, alluding to previous studies.
Response:
We thank the reviewer for the comment. We have restructured and renamed the conclusion section, and added the following:
- a contrast to the paper by Dumitru et al. (2019) in section 5.2;
- an extension on the Latapí Agudelo (2020) in section 6.2;
- a reflection on the Higgins (2020) in section 7;
- various other related reflections on other studies in the discussion and conclusion sections.
Moreover, Section 5.1, 6.1, and 7 all provide contrasts, reflections, and extensions to prior research.
Comment 8:
The format of the references should be reviewed.
Response:
We thank the reviewer for the comment.. The format conforms to what is suggested by the journal. There were a few inconsistencies which have been rectified. We have once again cross checked.
-----
Overall, we sincerely thank all three reviewers for their insightful feedback. We have applied their constructive suggestions for improving the paper’s structure, argumentation, and clarity.
Sincerely,
The Authors
-------------------------------------------------------------------------------------
Reviewer 3 Report
Figure 1 is not clear for understand, it should be corrected.
In this paper is not clear described the sustainability dictionary measure developed in the study. I really did not know researched words, what could be big impact for future research. Where is the developed sustainability dictionary (words from it), what was tested? Transparency measurement, using in this OPEC and non-OPEC evaluation (processing) for me is not very understandable.
Results are very interesting and clear described.
There is not so much recent sources
Author Response
Reviewer 3:
Comment 1: Figure 1 is not clear for understand, it should be corrected.
Response: We thank the reviewer for the comment. This was a concern of Reviewer 1 also and we have now removed Figure 1 and instead included an itemized summary of steps in the research to provide better clarity.
Comment 2: In this paper is not clear described the sustainability dictionary measure developed in the study. I really did not know researched words, what could be big impact for future research. Where is the developed sustainability dictionary (words from it), what was tested?
Response: This is a 668-word dictionary that has been rated by experts and we are currently working on a research publication based on other potential applications of the dictionary. An outcome of that effort will be the publication of the dictionary online. Since this journal allows online addendums, we'll add this information when the dictionary is published online.
Comment 3: Transparency measurement, using in this OPEC and non-OPEC evaluation (processing) for me is not very understandable.
Response: We have clarified the explanation as follows:
- In section 4.2.1 the corresponding calculations of the transparency measure for each corporate report has been linked to the definitions in Section 3.1.3.
- In section 4.2.3 the comparison of OPEC with non-OPEC has now been clearly marked as four steps with subtitles included.
Comment 4: Results are very interesting and clear described.
Response: We thank the reviewer for the comment.
Comment 5: There is not so much recent sources
Response: This was a common concern for all three reviewers, and this has now been addressed by adding 12 additional references, 10 of which are most recent and also inclusive of references from the Sustainability journal. These new references cover topics such as corporate information disclosure, those specific to the energy sector, analysis of corporate reports through content analysis and text mining and entropy as a measure.
-------------------------------------------------------------------------------------
Overall, we sincerely thank all three reviewers for their insightful feedback. We have applied their constructive suggestions for improving the paper’s structure, argumentation, and clarity.
Sincerely,
The Authors
Round 2
Reviewer 1 Report
The paper is substantially improved, but following aspects are still needed to be clarified.
Comment 5:
I understood the explanations, but I still think it would be useful for readers to to create an appendix for the first four sources, to know what they are:
- 30 most-cited seminal papers in the topic of sustainability;
- 10 most relevant and most cited books on the topic of sustainability;
- the past five years of sustainability reports of companies ranked as most sustainable - or how many companies are most sustainable;
- 30 most influential non-governmental organizations (NGO)
Comment 6:
I did not find the sentence below in the text.
Reports published in the years 2015-2017 were used for the research (Section 3.1.1).
Comment 8:
The following text, from the response to the comment, could be added to the conclusions to justify the fact that you stopped in 2017 as the analysis period.
Additional point to note is that significant restructuring happened in the energy industry (for instance the constitution of OPEC and non-OPEC groups) post 2017. This also significantly impacts the homogeneity of the dataset if we were to include more recent reports.
Author Response
Reviewer 1:
Comment 5:
I understood the explanations, but I still think it would be useful for readers to to create an appendix for the first four sources, to know what they are:
Response: We thank the reviewer for this suggestion. As mentioned in our earlier response our intention was to publish very detailed information related to the corpus in a paper dedicated to building and applying a sustainability dictionary that is currently under preparation. However, we respect the reviewer's suggestion and we have now added this information to the Appendix. Note that a larger pool of data was used for the original source corpus which was whittled down based on results of data validation.
We have also modified the narrative in 3.1.1 to more accurately reflect the description of these sources.
(i) 30 well-cited seminal papers in the topic of sustainability; (ii) ten relevant books on the topic of sustainability; (iii) a selection of the prior five years of sustainability reports of companies ranked by Forbes as most sustainable; (iv) content extracted from a selection of the thirty most influential non-Governmental or-ganizations (NGO) that play a key role in promoting sustainability practices; and (v) the most recent 1000 tweets of each of these NGOs. Details of the sources of this data is available in Appendix 1.
Comment 6:
I did not find the sentence below in the text -Reports published in the years 2015-2017 were used for the research (Section 3.1.1).
Response: We added a different sentence in 4.2.1 (not 3.1.1) that we thought was equivalent –"Data was compiled in the summer of 2018 for the prior three years." Now we have changed it to the above statement as suggested by the reviewer.
Comment 8:
The following text, from the response to the comment, could be added to the conclusions to justify the fact that you stopped in 2017 as the analysis period - "Additional point to note is that significant restructuring happened in the energy industry (for instance the constitution of OPEC and non-OPEC groups) post 2017. This also significantly impacts the homogeneity of the dataset if we were to include more recent reports in this study."
Response: We thank the reviewer for this useful suggestion. This statement has now been added to Section 4.2.1.
Reviewer 2 Report
The authors have made all suggested changes
Author Response
Comment:
The authors have made all suggested changes.
Response:
We thank the reviewer accepting our responses. We have re-read the article and also done a spell check. We have cross checked the references again for some minor issues.